# Analysis of Feeding Behavior Characteristics in the Cu/Zn Superoxide Dismutase 1 (SOD1) SOD1G93A Mice Model for Amyotrophic Lateral Sclerosis (ALS)

**DOI:** 10.3390/nu15071651

**Published:** 2023-03-28

**Authors:** Yoshihiro Kitaoka, Soju Seki, Sou Kawata, Akira Nishiura, Kohei Kawamura, Shin-ichiro Hiraoka, Mikihiko Kogo, Susumu Tanaka

**Affiliations:** First Department of Oral and Maxillofacial Surgery, Graduate School of Dentistry, Osaka University, 1-8 Yamadaoka, Suita 565-0871, Japan; d.yoshi.k.hiro@gmail.com (Y.K.); s.t.1955224@gmail.com (S.K.); fm.kariage@gmail.com (A.N.); kohei.kawamu@gmail.com (K.K.); hiraoka.shin-ichiro.dent@osaka-u.ac.jp (S.-i.H.); kogo.mikihiko.dent@osaka-u.ac.jp (M.K.); tanaka.susumu.dent@osaka-u.ac.jp (S.T.)

**Keywords:** amyotrophic lateral sclerosis (ALS), SOD1G93A, ALS mouse model, feeding disorders, mastication movements, single-shot multibox detector (SSD), mesencephalic trigeminal neurons (MesV), patch clamp, primary sensory neurons

## Abstract

Amyotrophic lateral sclerosis (ALS) is a progressive disease affecting upper and lower motor neurons. Feeding disorders are observed in patients with ALS. The mastication movements and their systemic effects in patients with ALS with feeding disorders remain unclear. Currently, there is no effective treatment for ALS. However, it has been suggested that treating feeding disorders and improving nutritional status may prolong the lives of patients with ALS. Therefore, this study elucidates feeding disorders observed in patients with ALS and future therapeutic agents. We conducted a temporal observation of feeding behavior and mastication movements using an open-closed mouth evaluation artificial intelligence (AI) model in an ALS mouse model. Furthermore, to determine the cause of masticatory rhythm modulation, we conducted electrophysiological analyses of mesencephalic trigeminal neurons (MesV). Here, we observed the modulation of masticatory rhythm with a prolonged open phase in the ALS mouse model from the age of 12 weeks. A decreased body weight was observed simultaneously, indicating a correlation between the prolongation of the open phase and the decrease observed. We found that the percentage of firing MesV was markedly decreased. This study partially clarifies the role of feeding disorders in ALS.

## 1. Introduction

Amyotrophic lateral sclerosis (ALS) is a progressive disease of the upper and lower motor neurons with an incidence of 2 per 100,000 people [1] that causes a myriad of symptoms, including paralysis with motor neuron damage, muscle weakness in limbs, dyspnea, and difficulty in eating and swallowing, leading to death within one to five years of onset [2,3]. Currently, ALS therapies are based on symptom management and respiratory support. Widely used approved drugs include riluzole, which protects neurons by inhibiting excitatory amino acid receptors, and edaravone, which inhibits oxidative damage to neurons by reducing lipid peroxidation through free-radical scavenging. However, these treatments only prolong life for a few months [4,5]. Although recent studies have used gene therapy in an attempt to treat ALS [6], an effective therapeutic agent to prevent ALS progression has not yet been developed. Moreover, there is currently no standard ALS treatment. Dysphagia is a frequent symptom in patients with ALS, and its early stage is mainly caused by oral dysfunction. Improving feeding and swallowing difficulties may improve the prognosis of ALS [7,8]. Therefore, it is necessary to elucidate feeding disorders in ALS and to study the relationship between eating disorders and disease progression. However, there are no precise biomarkers for early diagnosis of ALS, and it is challenging to observe patients with ALS from the early to terminal stages of the disease. Therefore, many studies on ALS disease progression have been conducted using ALS mouse models. Cu/Zn Superoxide dismutase 1 (SOD1) is the most frequently mutated gene in familial patients with ALS [9,10]. Hence, SOD1G93A transgenic mice are the most commonly used model in ALS studies [10,11]. These ALS mice develop symptoms such as hindlimb twitching and wobbling, followed by progressive muscle stiffness in the limbs and death at approximately 20 weeks of age [12,13,14,15]. Furthermore, it has been observed that changes in body weight occur as the disease progresses [16]. It has been reported that metabolic abnormalities occur as ALS progresses that may contribute to weight loss [17]. However, after the age of 9 weeks, ALS mice show changes in forelimb and hindlimb grip strength but no effect is observed on locomotion in the terminal phase [16]. After the age of 13 weeks, ALS mice show a rapid decrease in their daily running distance [18]. Hence, there is currently no clear understanding on the relationship between locomotion and body weight in ALS mice. Furthermore, the relationship between feeding behavior, mastication movements, and body weight in ALS mice is unclear. Therefore, in this study, we used an ALS mouse model to elucidate the characteristics of mastication movement and feeding behavior. Moreover, we examined the relationship between mastication movement and body weight during the disease progression.

Different methods exist for recording feeding behavior in ALS mice models [19,20]. Among them, videography is a non-invasive method of observing animal behavior. Observations of mastication movements include a mouse jaw movement tracking system that reconstructs 3D movement trajectories at arbitrary points on the mandible [21]. Moreover, there are reports of using a high-speed camera to efficiently record mouse jaw movements at a high speed of approximately 5 Hz [21]. However, both methods require suppression of jaw movements under nonphysiological conditions. The mastication cycle in mice is classified into three phases: open mouth, closed mouth, and occlusal phase. The characteristic mastication movements in mice are mainly formed by the coordinated movements of the masseter and temporal muscles [21,22]. Lever et al. used a video camera to capture the mastication movements of ALS mice and visually judged the opening and closing of the mouth using 30 images per second to examine the mastication cycle [23]. Therefore, in this study, we analyzed the mastication cycle of mice using a video camera to capture the mastication movement of mice and determine the opening and closing of the mouth using 25 images per second.

Artificial Intelligence (AI), including deep learning, has developed rapidly in recent years. It is now easier to code than before because computers automatically learn how to extract useful features [24]. By substituting AI for the experimental analysis previously performed, vast amounts of information can be analyzed quickly [25,26]. In medicine, there have been many attempts to apply AI to analyze various images, including radiographs, magnetic resonance, and ultrasound images [27]. Single-shot Multibox Detector (SSD) is a state-of-the-art algorithm based on deep learning techniques for object detection from images [28]. SSD uses a deep convolutional neural network (CNN) consisting of more than sixteen layers and is one of the high-performance models for AI systems in image recognition [28,29,30]. In medicine, the system has been used to detect esophageal cancer, and hyoid bone in images from swallow contrast studies, animals in motion, and assess swallowing [31,32]. In this study, we used SSD to quantitatively evaluate the characteristics of feeding behavior in ALS mice and develop an AI model that automatically detects mouth opening and closing during mastication.

Masticatory movements are controlled by masticatory muscles, trigeminal motor neurons (MoV), and mesencephalic trigeminal neurons (MesV), which are primary sensory neurons. Recent studies in mice and rats have reported that the MoV and the MesV play an essential role in the formation of masticatory rhythm [33]. MesV has been considered as constituting a sensory nerve associated with stretch reflexes by projecting sensory stimuli from the muscle spindles of the mouth-closing muscles to the hypothalamus. However, it plays a role in transmitting movement-related intrinsic sensory information to other regions of the central nervous system (CNS) [34,35]. Electrophysiological investigations in the MoV of neonatal ALS model mice showed overstimulation, as also observed in other motor neurons [36]. Irregular firing activity with firing inhibition has been observed in the MesV of neonatal ALS mice, attributed to the reduction of Nav1.6 type Na^+^ current [37]. These early changes in neuromuscular transmission in ALS mice are speculated to occur prior to the onset of ALS-related symptoms [38]. Electrophysiological studies show different characteristic changes in MoV and MesV. This suggests that not only motor neuron abnormalities, which have been considered as the cause of ALS and have been mainly studied, but also primary sensory neuron abnormalities, which have received attention, must be investigated in the pathogenesis of ALS. However, there are no reports of electrophysiological characteristics in the mature MesV despite reports of significant abnormalities in the neonatal MesV in the ALS mouse model. Additionally, the presence or absence of anatomical and electrophysiological abnormalities in the MesV after disease progression remains unknown. In this study, we investigate the electrophysiology of MesV using the whole-cell patch clamp technique to elucidate the characteristics of the masticatory motor center in the mature ALS model mice. Elucidating the modulation of mastication rhythm and the abnormalities of primary sensory neurons that control mastication may improve eating disorders in patients with ALS in the future. Furthermore, it may lead to the development of new ALS drugs targeting primary sensory neurons.

## 2. Materials and Methods

### 2.1. Experimental Animals

ALS mice model (JAX strain: C57B6SJL-Tg) (SOD1G93A)1Gur/J) (mSOD1 mice) (Jackson Laboratory: Bar Harbor, MA, USA) and their wild-type mice (C57B6J) (WT mice) were used in this study [37]. All animal protocols were conducted in accordance with the “Guidelines for the Proper Conduct of Animal Experiments 2006” (Guidelines) established by the Science Council of Japan and approved by the Safety Committee for Genetic Recombination Experiments at Osaka University (Approval No. 04538 of the Safety Committee for Genetic Recombination Experiments) and the Animal Experiment Committee of the Graduate School of Dentistry, Osaka University (The approval of the Safety Committee for Genetic Recombination Experiments of Osaka University (Approval No.: 04538) and the Animal Experiment Committee of the Graduate School of Dentistry, Osaka University (Approval No.: Animal Experiment Committee: Dent-20-005-0) were obtained. In this experiment, 7 to 18-week-old mice (male, mSOD1; *n* = 10, WT; *n* = 9) were used for feeding behavior studies including AI model development, and 12-week-old mice (male, mSOD1; *n* = 5, WT; *n* = 5) were used for electrophysiological studies of MesV (Table 1). Genotypes of mice were determined via RT-PCR using tail samples. Each mouse was housed in an individual cage in a controlled environment with a temperature of 23 °C, 60% humidity, and light and dark periods every 12 h and fed ad libitum on solid feed (MF: Oriental Yeast, Tokyo, Japan) [39].

### 2.2. Video Recording of Opening and Closing Mouth Movements in ALS Model Mice

A cheese feed for small animals (cut cheese: Doggy Man Hayashi, Osaka, Japan) was fixed to the side wall of the cage. No significant difference was found in the mastication cycle of mice chewing soft foods compared to that of mice chewing regular pellets [40] and with the aim of recording a large number of feeding behaviors of mice within a certain time period, we used cheese feed in this study [41,42]. Two video cameras (HANDYCAM, model HDR-XR520V: Sony Corporation, Tokyo, Japan; Everio, model GZ-HM890: JVC Corporation, Yokohama, Japan) were used for 30 min of recording (25 images/s) [23] (Figure 1a) to observe the feeding behavior of mice from two lateral directions (115 and 165 mm from the solid feed, respectively). Spontaneous feeding behavior was recorded without prior food restriction. Since other methods using high-speed cameras [21] require jaw movements to be controlled under nonphysiological conditions, this study employed a protocol that allows jaw movements to be observed under physiological conditions [23].

### 2.3. Development of an AI Model for Detecting Opening and Closing Mouth Movements in Mice

In developing an AI model for detecting the opening and closing mouth movements of mice using deep learning techniques, we used the Single Shot Multibox Detector (SSD), a general object recognition algorithm [28]. The criteria for the “open” mouth condition were as follows: the upper and lower jaws were vertically at their maximum opening and closing mouth images of the mouse’s lateral face, and the outlined line from the nasal tip of the lateral face to the mandible’s lower edge was not continuous on the image (Figure 1b). The criteria for the “close” mouth condition were as follows: completely closed mouth with occlusion of the upper and lower jaws and continuity of the contour line from the nasal edge of the lateral aspect to the mandible’s inferior border on the image (Figure 1c) [23]. This allowed us to classify more than 99% of the images as “open” or “close”. A total of 3009 (open, 1504 images; close, 1505 images) from 10 mice were used to develop the opening and closing mouth motion detection AI (Figure 2a).

Of these, 2036 images from 7 mice were used for SSD training, and 873 images were used for SSD validation (Figure 2a). One hundred images generated from three mice, which were not used in training or validation, were used to evaluate SSD (Figure 2a). Video recordings of feeding behavior over time were used to identify and detect “open” and “close” using the AI model. The detection screen displayed a confidence score (Figure 2c), which is the probability of being confident that the rectangle contains the predefined information. The highest confidence score was 1.0, confirming that “open” and “close” conditions were appropriately detected by SSD (Figure 2c). Based on the percentage of agreement with the SSD detection results, the AI model was evaluated using 100 images (“open”; 50 images, “close”; 50 images in advance). The AI findings agreed by 88% and 92% for the “open” and “close” images, respectively (Figure 2b). To determine the opening and closing mouth movements using SSD, two scenes were extracted; one from the first 15 min of a 30-min video recording of feeding behavior and another scene from the last 15 min [23]. The AI model evaluated “open” or “close” for 25 images per second (0.04 s per image) and found multiple consecutive “open” judgments in response to the opening mouth movements and multiple consecutive “close” judgments in response to the closing mouth movements (Figure 2c). Based on these results, “open phase duration”, “close phase duration”, and “open and close ratio” were set as parameters of jaw movement, and measurements were made over time. The “open phase duration” was defined as 0.04 s × (the number of images consecutively identified as “open”) (Figure 2c). The “close phase duration” was defined as 0.04 s × (the number of images consecutively identified as “close”) (Figure 2c). The “open and close ratio” was defined as the ratio of “open” to “close” in consecutive opening and closing mouth movements. Each jaw movement parameter was measured over time.

### 2.4. Observation of Feeding Behavior and Body Weight Measurement of ALS Model Mice

During the observation period, all mice were allowed to feed freely. We weighed the mice prior to beginning the observation of their feeding behavior. Video recording was conducted once a week from 18:00 to 22:00 for 30 min. Based on the methodology reported by Ushimura et al. and Kida et al., we used the following feeding behavior parameters: “body weight”, “feeding onset time”, “total feeding time”, and “total food intake” [39,43]. The feeding onset time was the time taken from the start of observation to the start of feeding behavior. The total feeding time was taken as the time spent only on chewing the feed, irrespective of other feeding behaviors observed during the 30 min after the start of the observation. The total food intake was defined as the change in the weight of the feed between the beginning and the end of the 30-min observation [39,43]. These parameters were measured weekly from 7 to 18 weeks of age, and the mSOD1 and WT groups were compared over time.

### 2.5. Electrophysiological Investigation of MesV in ALS Mice Model

WT and mSOD1 mice at 12 weeks of age were used in this study. Following previously reported methods [44,45,46,47,48,49], mice were administered isoflurane to cause general anesthesia, decapitated, and cooled in artificial cerebrospinal fluid for cutting (cutting composition; 126 mM NaCl, 3 mM KCl, 1.25 mM NaH_2_PO_4_, 26 NaHCO_3_, 10 mM glucose, 1 mM CaCl_2_, 5 mM MgCl_2_, 4 mM lactic acid). The brain stem tissue was removed from the skull. After the brainstem block was prepared, it was fixed to a slice-preparation device using an ultra-low melting point agarose gel (agarose type VII: Sigma-Aldrich, St. Louis, MO, USA) to prepare a 300 μm thick coronal brainstem slice specimen containing MesV. The prepared slice specimens were transferred to standard artificial cerebrospinal fluid normal-Artificial cerebrospinal fluid (ACSF) (N-ACSF. Composition; 124 mM NaCl, 3 mM KCl, 1.25 mM NaH_2_PO_4_, 26 NaHCO_3_, 10 mM glucose, 2 mM CaCl_2_, 2 mM MgCl_2_) for recording at room temperature (22–24 °C). The samples were allowed to stand for 40–50 min and then used for recordings. All extracellular fluids were mixed with a 95% O_2_ + 5%CO_2_ gas mixture.

Brain stem slices were placed in a recording chamber (acrylic, 2.0 mL volume) on the stage of an upright Nomarski infrared differential interference microscope (BX51W1: Olympus, Tokyo, Japan) and fixed using slice anchors (Warner Instruments: Holliston, MA, USA). MesV, pseudo-monopolar round cells of approximately 20–40 μm in diameter, were anatomically identified under infrared fluoroscopic conditions while irrigating with N-ACSF at a rate of 2 mL/min. Whole-cell patch clamp recording was performed after gigaseal formation. Recording electrodes were prepared using a microglass tube electrode generator (P-87: Sutter Instruments, Novato, CA, USA) with a 1.5 mm OD and 1.12 mm ID borosilicate glass tube (150 EA/PKG, InterMedical: Tokyo, Japan). The electrode was prepared so that the tip resistance was 3–5 MΩ. The electrode inner solution was composed mainly of potassium salts: 115 mM K-gluconate, 25 mM KCl, 9 mM NaCl, 10 mM HEPES, 0.2 mM EGTA, 1 mM MgCl_2_, 3 mM K2-ATP, 1 mM Na-GTP, and adjusted to pH 7.25. Only neurons with an access resistance of less than 15 MΩ that formed between the neuronal surface and the electrode were included in the recording. The electrical activity of MesV was recorded under voltage-clamp (v-clamp) or current-clamp (c-clamp) conditions. Electrical signals were amplified with a patch clamp amplifier (Multiclamp 700B: Molecular Devices, San Jose, CA, USA) and digitally converted with an analog-to-digital converter (Digidata 1550 A: Molecular Devices, San Jose, CA, USA). The signals were recorded and analyzed using a personal computer and dedicated software (CLAMPEX 10.6: Molecular Devices, San Jose, CA, USA).

### 2.6. Statistical Analysis

Statistical analyses were performed using Microsoft Excel (Microsoft, Redmond, WA, USA), and SPSS 24.0 statistics program (SPSS Inc: Chicago, IL, USA). The Kolmogorov–Smirnov or Shapiro–Wilk test was used for normality, and Levene’s test was used for equality of variances. Independent subsets were analyzed with Student’s *t*-test. Cohen’s effect size (d) was calculated for pairwise comparisons. All data are expressed as mean ± standard error (S.E.). Comparisons between the WT and mSOD1 groups in the “open” and “close” phase duration, body weight, feeding onset time, total feeding time, and total food intake were evaluated using two-way repeated measures ANOVA. If an interaction was detected between the two groups, post-hoc comparisons were performed with a Student’s *t*-test. The relationship between the opening and closing mouth phase time and body weight was evaluated using Pearson’s product ratio correlation coefficient test. The significance level was set at 5%.

## 3. Results

### 3.1. AI Model for Opening and Closing Mouth Movement in Mice

The AI model detected the opening and closing mouth movements. The open phase duration, close phase duration, and open and close ratio were measured and compared between the WT and mSOD1 groups at ages 7 to 18 weeks. The mSOD1 group showed an increase in open phase duration after 12 weeks of age and a significant increase after 17 weeks of age (F(1, 146) = 1.95, *p* = 0.01, d = 1.24, 17 weeks; WT, 0.05 ± 0.01 s, mSOD1, 0.08 ± 0.01 s, *p* = 0.01, d = 1.69, 18 weeks; WT, 0.05 ± 0.01 s, mSOD1, 0.09 ± 0.01 s, *p* = 0.01, repeated-measured ANOVA followed by student-*t* post hoc comparison, Figure 3a). Neither close phase duration (F(1, 182) = 1.04, *p* = 0.99, repeated-measured ANOVA, Figure 3b) nor open and close ratio (F(1, 263) = 1.09, *p* = 0.12, repeated-measured ANOVA, Figure 3c) from 7 to 18 weeks of age were not significant between the two groups.

### 3.2. Changes in Body Weight and Feeding Behavior in the ALS Mice Model over Time

The normalized weight of the WT group increased from 7 to 12 weeks of age, while that in the mSOD1 group decreased after 12 weeks of age. A significant decrease at 15, 17, and 18 weeks (F(1, 218) = 0.88, *p* = 0.01, 15 weeks; WT, 1.23 ± 0.03, mSOD1, 1.14 ± 0.04, *p* = 0.01, d = 0.72, 17 weeks; WT, 1.25 ± 0.05, mSOD1, 1.08 ± 0.05, *p* = 0.01, d = 1.06, 18 weeks; WT, 1.31 ± 0.04, mSOD1, 1.05 ± 0.05, *p* = 0.01, d = 1.61, repeated-measured ANOVA followed by student-*t* post hoc comparison, Figure 4a) was noted. Both groups showed a decreasing trend in feeding onset time, but there was no significant difference between the two groups (F(1, 164) = 1.09, *p* = 0.56, 8 weeks; WT, 323.50 ± 84.31 s, mSOD1, 426.50 ± 294.50 s, 18 weeks; WT, 58.83 ± 17.60 s, mSOD1, 56.00 ± 8.00 s, repeated-measured ANOVA followed by student-*t* post hoc comparison, Figure 4b). The total feeding time did not change from 7 to 18 weeks of age in the WT group but showed an increasing trend in the mSOD1 group (F(1, 163) = 0.79, *p* = 0.99, 17 weeks; WT, 180.80 ± 32.03 s, mSOD1, 303.14 ± 39.48, 18 weeks; WT, 89.00 ± 19.49 s, mSOD1, 417.50 ± 151.67 s, repeated-measured ANOVA followed by student-*t* post hoc comparison, Figure 4c). Total food intake showed an increasing trend in both groups but no significant difference between them (F(1, 169) = 1.20, *p* = 0.23, 7 weeks; WT, 0.08 ± 0.02 g, mSOD1, 0.02 ± 0.01 g, 18 weeks; WT, 0.16 ± 0.03 s, mSOD1, 0.27 ± 0.09, repeated-measured ANOVA followed by student-*t* post hoc comparison, Figure 4d).

### 3.3. Correlation between the Open and Close Phase Duration and Body Weight in ALS Mice Model after 12 Weeks of Age

To examine factors related to weight loss, we investigated the correlations between normalized weight and open and close phase duration, and open and close ratio after 12 weeks of age. No correlation was found between normalized weight and open phase duration in the WT group (r = 0.08, *p* = 0.57, d = 1.32, Figure 5a), but a negative correlation was found in the mSOD1 group (r = −0.44, *p* = 0.01, d = 1.29, Figure 5b). No correlation was found between normalized weight and close phase duration in either WT or mSOD1 group (WT; r = 0.22, *p* = 0.19, d = 1.20, mSOD1; r = −0.07, *p* = 0.67, d = 1.40, Figure 5c). No correlation was found in the normalized weight and open and close ratio for the WT and mSOD1 groups (WT; r = 0.13, *p* = 0.41, d = 1.21, mSOD1; r = 0.06, *p* = 0.70, d = 1.34, Figure 5d).

### 3.4. Electrophysiological Characteristic Modulation of MesV in 12-Week-Old ALS Mice Model

We compared the electrophysiological properties of MesV in 12-week-old mice (Figure 6a). Under v-clamp conditions in basic membrane properties, the resting membrane potential (RMP) of the mSOD1 group was significantly lower than that of the WT group (F(1, 23) = 0.39, WT: −52.50 ± 4.50 mV, mSOD1; −84.80 ± 5.70 mV, *p* = 0.01, d = 1.27, Table 2). No significant differences in other basic membrane properties were observed (Table 2).

When MesV were applied to square-wave depolarizing pulses under c-clamp conditions, they were divided into two types of neurons; one showing a single action potential and the other showing continuous firing activity (Figure 6a). The mSOD1 group showed a decreasing trend in the proportion of MesV with continuous firing activity (WT; 56%, 5/9, mSOD1; 20%, 3/15, *p* = 0.07, Figure 6b). The action potential (AP) was observed using 10 pA, 10 ms square wave depolarizing pulse under c-clamp (Figure 6c). No significant differences in AP characteristics (Spike height, Half width, Slope, AHP peak and duration) were observed between the mSOD1 and WT groups (Figure 6c, Table 3).

Continuous firing activity was observed by a square-wave hyperpolarizing pulse of 0–1000 pA and a 500 ms square wave depolarizing pulse under c-clamp (Figure 6d). In MesV showing continuous firing activity, firing duration in the mSOD1 group was significantly shorter than that in the WT group (F(1, 6) = 39.69, WT; −55.50 ± 4.50 ms, mSOD1; −23.80 ± 5.70 ms, *p* = 0.01, Figure 6e). In frequency–current curves (F-I curves), MesV with continuous firing activity in the mSOD1 group showed a significant increase in frequency (F(1, 48) = 3.85, *p* = 0.01, 700 pA; WT, 124.80 ± 12.10 Hz, mSOD1; 157.90 ± 12.80 Hz, *p* = 0.02, d = 1.49, 800 pA; WT, 129.30 ± 12.40 Hz, mSOD1, 174.10 ± 13.50 Hz, *p* = 0.01, d = 1.66, 900 pA; WT, 136.90 ± 11.10 Hz, mSOD1, 184.30 ± 21.70 Hz, *p* = 0.03, d = 1.59, 1000 pA; WT, 143.10 ± 9.60 Hz, mSOD1, 194.03 ± 18.70 Hz, *p* = 0.02, d = 1.69, Figure 6f).

## 4. Discussion

This study observed the modulation of masticatory rhythm and weight loss in an ALS mouse model. Electrophysiological examination of the MesV in adult mice revealed a decrease in the percentage of firing neurons and modulation of firing activity. To our knowledge, this study is the first to indicate that the characteristic changes of MesV reported in the neonatal ALS mouse model also occur in adult ALS mice.

### 4.1. Mastication Movement Analysis of ALS Model Mice Using AI

We measured mastication movements noninvasively by recording the feeding behavior of mice using a video camera. Furthermore, by developing an opening and closing mouth motion detection AI model that automatically detects the opening and closing mouth, it became possible to process a large amount of image data. The agreement rate between the AI evaluations and those made by the researchers for the opening and closing mouth of the mouse was 88% and 92%, respectively, showing a high agreement rate. Furthermore, the open phase duration of ALS model mice measured by the AI model ranged from 0.06 to 0.12 s, and the close phase duration ranged from 0.06 to 0.08 s. The opening and closing mouth motion cycles of mice reported by Lever et al. and Yoshida et al. were approximately 5 Hz that were close to the results of this study [23,50]. We consider the AI model appropriate for investigating feeding behavior and mastication rhyme. However, it has several limitations. First, the images used to train the opening and closing mouth are selected visually by the researcher, which may lead to variations in the results of opening and closing mouth decisions owing to differences in the judgment criteria among researchers. Therefore, in this study, the fully open and fully close images selected by two researchers were used to train the AI model to ensure minimal variation in the results. Therefore, the number of images used is fairly small, and the characteristics of the AI model are not yet clear. Accumulating a large number of datasets to build a highly accurate CNN may not improve accuracy, and the selection of datasets in future research using AI models should be considered in detail according to the need to improve model performance [51]. The selection of datasets in future studies using AI models should be extensively considered to improve the model performance [51]. 

The mouse AI model developed in this study can be replicated to mimic a human AI model for opening and closing mouth motion detection. Human mastication movement detection AI can non-invasively detect the modulation of mastication rhythm in real-time and can be used in home care and to determine disease progression in patients with ALS.

### 4.2. Relationship between Modulation of Masticatory Movement and Body Weight in ALS Mice Model

Body weight loss was observed in ALS mice after 12 weeks of age. The total food intake of ALS model mice increased in proportion to the age of the mice. It did not differ significantly from that of wild-type mice. Therefore, it was inferred that the weight loss was observed owing to metabolic abnormalities as the disease progressed, as reported in the past [17]. The relationship between locomotion and body weight in the ALS mouse model has yet to be clearly established [16,18]. Here, the feeding onset time of ALS mice tended to decrease with increasing age and was not significantly different from that of wild-type mice. Therefore, no apparent reduction in locomotion was suspected in the ALS mouse model. Furthermore, regarding the relationship between mastication movements and weight loss, as mastication plays a major role in food intake and digestion, it is sought to influence body weight through a variety of mechanisms. Previous studies in humans have reported that increasing the number of times that food was chewed from 15 to 40 times, resulted in decreased food intake and increased satiety, suggesting that prolonged chewing exercise may influence weight loss through decreased eating speed [52]. In this study, the average duration of mastication was 167.3 s in 12-week-old ALS mice and 417.5 s in 18-week-old mice, whereby the modulation of mastication rhythm and weight loss were observed. This suggested that the prolonged mastication time may have contributed to the weight loss through the decreased feeding rate.

### 4.3. Changes in MesV Properties in Mature ALS Model Mice

ALS mice began to lose weight at 12 weeks of age in this study. The correlation between weight loss and the modulation of opening movements was observed after 12 weeks of age. Therefore, an electrophysiological study was conducted in the 12-week-old ALS mice. This study observed a marked decrease in the resting membrane potential among the basic membrane properties in the MesV of mature ALS mice model. Slow inward rectifying electric current (I(h)) is involved in stabilizing the resting membrane potential and regulating cell excitability in MesV [47]. Moreover, it was reported that the membrane excitability and discharge characteristics of MesV could be significantly altered by regulating the sustained sodium current (I(NaP)), one of the Nav1.6-type Na^+^ currents [48]. 

Furthermore, I(NaP) is slowly inactivated and is not directly involved in the generation of transient action potentials [53], but is associated with the control of membrane excitability in the voltage region just below the threshold for spike generation [49]. These suggest that the decrease in the percentage of firing neurons observed in this study may be related to a decrease in I(NaP). Although continued firing activity induced under conditions of sustained depolarization of the membrane potential is observed in MesV during infancy [45,47,53], in this study, in the mature MesV, continuous firing activity was observed, but no endogenous burst firing activity was observed. Comparisons in firing neurons showed a significant increase in spike frequency and a decrease in firing duration in the MesV of mature ALS mice model. In addition to sodium current, previous reports have implicated transient outward current (I(TOC)) and 4-AP-sensitive persistent outward current (I(4-AP)) in the duration of firing activity in the MesV [44], especially I(4-AP), that determines whether neurons exhibit persistent firing or adapts in response to depolarizing currents [44]. A previous study reported that transient outward current (I(TOC)) and 4-AP-sensitive persistent outward current (I(4-AP)), in addition to sodium current, affected the duration of MesV firing activity. In particular, I(4-AP) can determine whether neurons exhibit sustained firing or respond or adapt to depolarizing currents [44]. The neurotransmitter modulation of this current and modulation of the resting membrane potential can alter the output properties of the MesV, and both sodium current enhancement and potassium current reduction induce overexcitation [35]. Thus, not only sodium conductance such as I(NaP), but also potassium conductance such as I(TOC) and I(4-AP), and I(h) may be modulated in the MesV of the mature ALS mice model In ALS mice, membrane excitability of not only motoneurons but also primary sensory neurons, MesV, may be progressively altered with modulation of multiple ion channels, including sodium and potassium conductance [54,55]. Although this study was conducted only under c-clamp conditions, previous studies on membrane excitability have examined the involvement of various currents under v-clamp conditions. Further investigation of the membrane excitability and discharge characteristics of MesV under v-clamp conditions needs to be carried out. Although ALS is caused by abnormalities in motor neurons, the finding of abnormalities in the characteristics of primary sensory neurons in ALS may not only help to clarify feeding disorders in ALS patients but may also lead to the development of new ALS drugs that target primary sensory neurons.

## 5. Conclusions

In this study, we investigated the characteristics of feeding behavior and electrophysiology of mesencephalic trigeminal neurons in an ALS mouse model. We observed a tendency for weight loss in ALS mice that was not attributed to a decrease in total food intake but a prolongation of feeding time, suggesting that the prolongation of feeding time with the modulation of mastication movement may have been a factor, as well as the metabolic abnormalities associated with the progression of ALS disease. In mesencephalic trigeminal neurons of ALS model mice, the characteristic changes reported in neonatal mice were also observed in mature mice. These results suggest that sodium conductance, as reported in neonatal ALS mice, and potassium conductance and other ion channels may be modulated. The results of this study partially clarify the relationship between feeding disorders and disease progression in ALS. Further studies remain warranted to elucidate the relationship between eating disorders and disease progression in ALS and to treat eating disorders in patients with ALS. The AI model developed in this study for detecting opening and closing mouth movements could not detect left and right mouth movements in mice. In the future, the goal is to develop a human opening and closing mouth detection AI, which will enable ALS patients to detect abnormalities in their opening and closing mouth movements by themselves at home, without having to visit the hospital. In the future, the human opening and closing mouth detection AI may become a non-invasive biomarker that is useful in determining disease progression and treatment efficacy in ALS patients.

## Figures and Tables

**Figure 1 nutrients-15-01651-f001:**
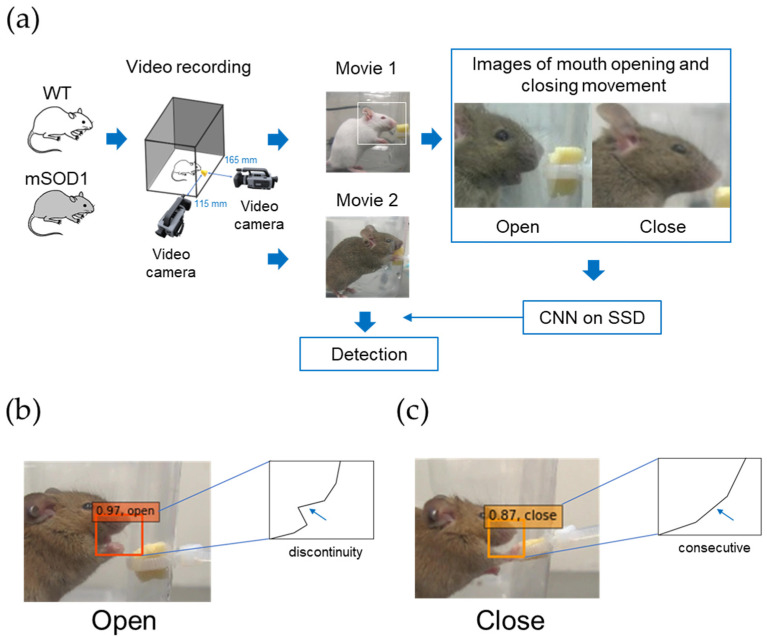
AI development for detecting the opening and closing mouth movements of mice. (**a**) Feeding behavior was recorded using two video cameras from two directions for 30 min. Images of mice’s mouths opening and closing were created from the video recordings and trained on the single-shot multibox detector (SSD). (**b**) Detection of “open” mouth in mice was considered so when the contour line from the nasal tip to the mandibular border of the lateral face was discontinuous on the image. (**c**) Detection of “close” mouth in mice, which was considered so when the condition in which the contour line from the nasal tip of the lateral aspect to the mandibular border was continuous on the image.

**Figure 2 nutrients-15-01651-f002:**
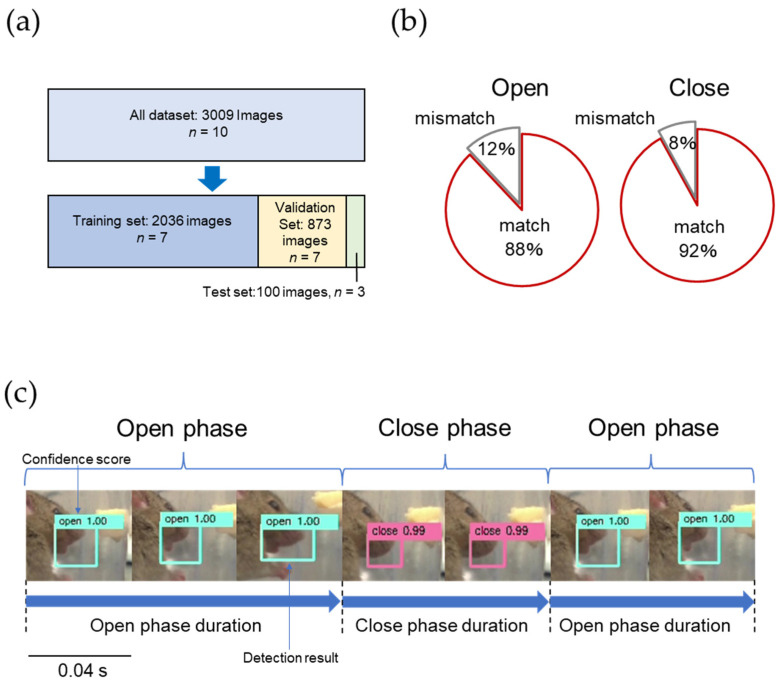
Detection of mouse opening and closing of the mouth. (**a**) Dataset used to develop the AI model for detecting mouth movements. (**b**) The percentage of agreement between the researchers and AI model in judging “open” and “close” mouth. (**c**) The measurement method of the open phase time and close phase time, 0.04 s × (the number of images judged as “open” or “close” consecutively), is open phase duration or close phase duration, respectively.

**Figure 3 nutrients-15-01651-f003:**
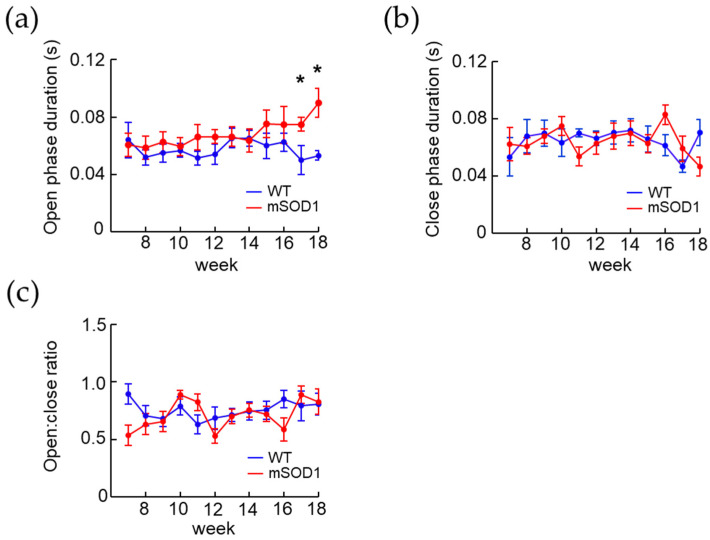
Changes over time in the opening and closing mouth characteristics in a mouse model of ALS. Open phase duration, close phase duration, and open:close ratio detected using an AI model. WT (mala, *n* = 10) and mSOD1 (male, *n* = 9) results from 7–18 weeks of age. (**a**) In mSOD1, open phase duration was prolonged after 12 weeks of age and significantly prolonged after 15 weeks old. (**b**,**c**) No significant difference was found for either close phase duration or open:close ratio. * *p* < 0.05.

**Figure 4 nutrients-15-01651-f004:**
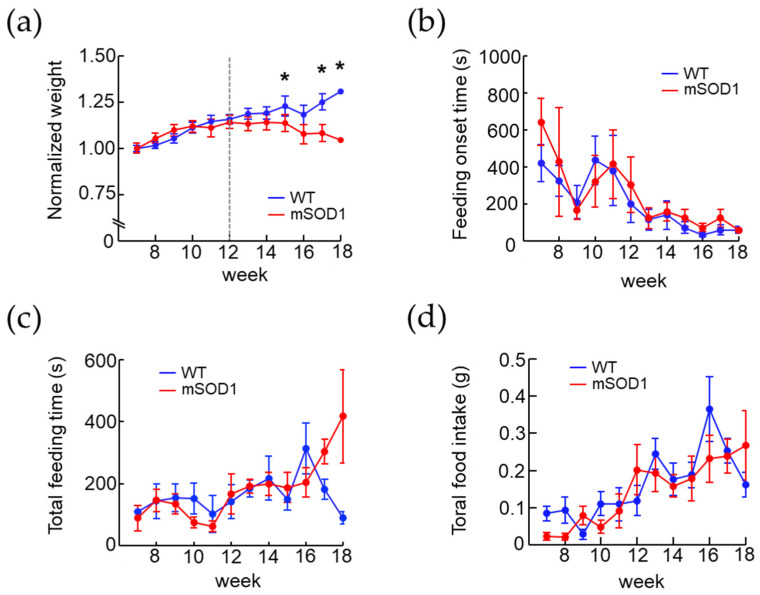
Changes over time in body weight and feeding behavior in a mouse model of ALS. (**a**) Normalized weight of mSOD1 peaks at 12 weeks of age and significantly decreased after age 14 weeks. (**b**) Normalized feeding time in 30-min observation period. Both groups showed increasing trend; however, no significant difference was found between them. (**c**,**d**) Normalized food intake and time taken to start feeding in a 30-min observation period. No significant differences were found. * *p* < 0.05.

**Figure 5 nutrients-15-01651-f005:**
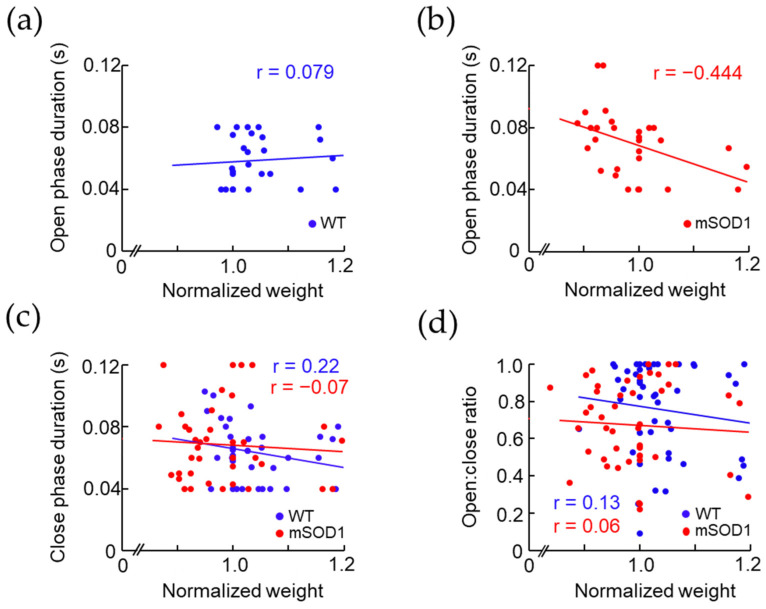
Correlation between body weight and the opening and closing mouth movements in ALS model mice after 12 weeks of age. The relationship between normalized body weight and open phase and close phase duration after 12 weeks of age is shown. (**a**,**b**) Despite no correlation between normalized body weight and open phase duration in WT, a negative correlation was observed in mSOD1. (**c**,**d**) No correlation between normalized body weight and close phase or open:close ratio was observed in WT and mSOD1.

**Figure 6 nutrients-15-01651-f006:**
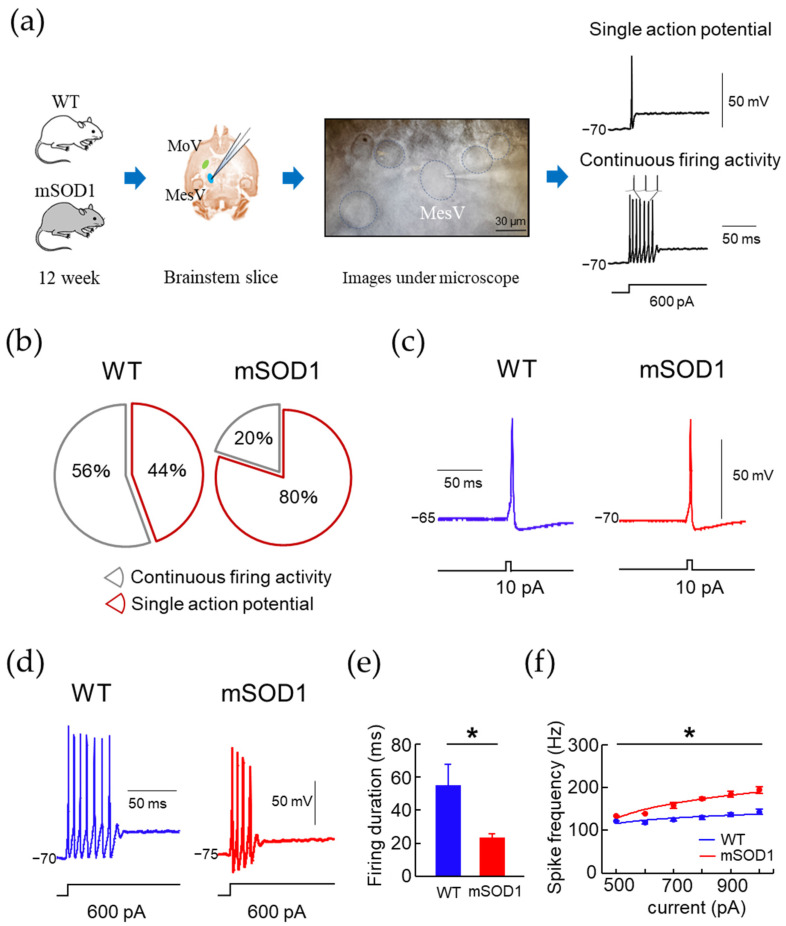
Whole-cell recording of MesV in a 12-week-old mouse model of ALS. (**a**) Schematic illustration of electrical recordings from MesV neuron showing “Single spiker” waveform and “Firing” waveform induced by a 600 pA square wave depolarizing pulse. (**b**) The “Single spiker” ratio was significantly higher in the mSOD1 group. (**c**,**d**) Action potential and continuous firing activity waveforms of WT and mSOD1 groups are shown. (**e**,**f**) Firing duration in the mSOD1 group was significantly shortened, while frequency in the 500–1000 pA injection current significantly increased. * *p* < 0.05.

**Table 1 nutrients-15-01651-t001:** Number of mice used in each experiment.

	WT	mSOD1
mouse	C57BL/6JJmsSLc	B6SJL-Tg (SOD1G93A)1Gur/J
Behavioral physiological Experiments	10	9
Electrophysiological Experiments	5	5
Total	15	14

**Table 2 nutrients-15-01651-t002:** Basic membrane properties of MesV in a 12-week-old mouse model of ALS.

	WT (*n* = 9)	mSOD1 (*n* = 15)
RMP, mV	−52.5 ± 4.5	−84.8 ± 5.7 *
R^in,^ MΩ	26.2 ± 5.8	21.3 ± 6.0
C^m,^ pF	82.1 ± 9.4	105.4 ± 9.6

Resting membrane potential (RMP) in the mSOD1 group was significantly lower, while input resistance (R_in_) and membrane capacitance (C_m_) were not significantly changed. * *p* < 0.05.

**Table 3 nutrients-15-01651-t003:** AP characteristics of MesV in a 12-week-old mouse model of ALS.

	WT (*n* = 9)	mSOD1 (*n* = 15)
AP		
Spike height, mV	79.1 ± 3.1	72.6 ± 2.4
Slope, mV/ms	−0.017 ± 0.006	−0.014 ± 0.007
Half width, ms	0.26 ± 0.04	0.24 ± 0.08
AHP		
AHP peak, mV	−8.8 ± 1.3	8.3 ± 0.5
duration, s	0.03 ± 0.01	0.04 ± 0.01

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
