# Peer review of "Analysis of Feeding Behavior Characteristics in the Cu/Zn Superoxide Dismutase 1 (SOD1) SOD1G93A Mice Model for Amyotrophic Lateral Sclerosis (ALS)"

_nutrients, 2023, doi:10.3390/nu15071651_

Round 1

Reviewer 1 Report

Abstract.

This section is sufficiently attractive to make the future reader feel interested enough to continue reading the rest of the article.

It clearly distinguishes the objective and how the experiment was conducted, the main variables studied, a general summary of the most important findings and the conclusion reached by the researchers: "This study partially clarifies the role of feeding disorders in Amyotrophic lateral sclerosis (ALS)".

1. Introduction

Line 36-37. It is true that our journal template allows truncation of syllables at the end of lines but that does not mean that it is no longer a spelling mistake. Please correct it manually and check all these cases throughout your paper. Thank you.

Lines 31-62. Good proofreading and good rhetorical questions/answers by the researchers. This makes the reading become a kind of virtual dialogue between the researchers and we readers become the first spectators..... Good work.

Lines 65-79. Interesting and up-to-date explanation of methods. Good work.

Lines 80-93. Very wise to include new methodological trends, such as Artificial Intelligence, to pursue research objectives. 

Line 97. I have the impression that there are two spaces after the point "neurons.  Recent". Please review and correct if necessary.

Ok, good work.

2. Materials and Methods

2.1 Experimental animals

Good explanation, good control, good permissions and respect for the ethics of scientific research.

Line 146. Table1. Please indicate clearly the number of total subjects (mice).

2.2 Video recording of opening and closing mouth movements in ALS model mice

Please justify why you have chosen this model (cut cheese: Doggy Man Hayashi, Osaka, Japan) and this recording protocol and not others and reference other papers that have also used them. This will give you credibility and reliability. Thank you

2.3 Development of an AI model for detecting opening and closing mouth movements in mice

Now you are right. Good explanation of why you have chosen Single Shot Multibox Detector.

2.4 Observation of feeding behaviour and body weight measurement of ALS model mice

Now yes. You explain very well why you have followed the methodology reported by Ushimura et al. and Kida et al.,

Good work.

2.5 Electrophysiological investigation of MesV in ALS mice model

Ok, good explanations

2.6 Statistical analysis

Please include here and later in the presentation of the results of the pairwise comparison (Student's t-test) the "effect size". Thank you

3. Results 

3.1 AI model for opening and closing mouth movement in mice

Please do not forget to include the effect size in the pairwise comparison.

3.2 Changes in body weight and feeding behaviour in the ALS mice model over time

Please do not forget to include the effect size in the pairwise comparison.

3.3 Correlation between the open and close phase duration and body weight in ALS mice model 315 after 12 weeks of age

Please do not forget to include the effect size in the pairwise comparison.

3.4 Electrophysiological characteristic modulation of MesV in 12-week-old ALS mice model

Please do not forget to include the effect size in the pairwise comparison.

Please indicate in the legend of table 2 what the asterisk (*) means.

4. Discussion

This is undoubtedly the best section of the whole paper and it is appreciated that the discussion has been structured in these two sections: "Relationship between modulation of masticatory movement and body weight in ALS mice model" and "Changes in MesV properties in mature ALS model mice", this allows the reader to frame in a more precise way the comparisons and relationships between previous literature and the findings detected in this research.

Good work

5. Conclusions

Although the initial objective is fulfilled and the contributions in this section are accurate, we would expect some explanation of the limitations detected in this study as well as the possible and future practical applications of the findings for application in human research.

Author Response

Dear Reviewer 1

Thank your for your thoughtful and appropriate comment.

We revised our manuscript according to your comment.

Best,

Soju Seki

Reviewer 2 Report

The authors have studied the feeding behavior and performed the electro-physiological study in the MesV neuron of the adult SOD1G93A mice.  The authors have also discussed the weakness of the study. My comments are provided below

1. The authors used the unbiased AIs to measure the feeding behavior and conclude that the feeding time of the SOD1 mice has increased although its a trend (line 301) and statistically not significant. The authors should either provide  p value or mention 'trend' in the sentence (line 465) so as not to mislead the reader.

2. The previous study has taken 30 images per second and this study has taken 25 images per second. The authors should discuss the relevance of this difference  if any.

3. The authors have performed the electro-physiological study on 12 week old mice and not on 14  or 16 or 18 weeks old where there is a significant weight difference. The authors should discuss about it.

Author Response

Dear Reviewer 2

Thank your for your thoughtful and appropriate comments.

I have revised our manuscript according to your comments.

Best,

Soju Seki

Reviewer 3 Report

Major concerns

  Why the electrophysiological properties of MesV were investigated in 12-week-old mice? After 12 weeks, ALS model mice and wt mice showed differences in body weight and open-close movement. 

Minor errors

Line 51 (Gurney et al. mice [10,11]. is “[10,11].”

Line 138 “viaRT-PCR ” is via RT-PCR

Line 250 MgCl2 is MgCl2

Line 394-399 two sentences were repeated.

Author Response

Dear Reviewer 3

Thank you for your thoughtful and appropriate comments.

I had revised our manuscript according to your comments.

Best,

Soju Seki
